# Efficacy and Safety of Mesenchymal Stem/Stromal Cell Therapy for Inflammatory Bowel Diseases: An Up-to-Date Systematic Review

**DOI:** 10.3390/biom11010082

**Published:** 2021-01-11

**Authors:** Jeffrey Zheng-Hsien Ko, Sheeva Johnson, Maneesh Dave

**Affiliations:** 1Department of Internal Medicine, Division of Gastroenterology and Hepatology, UC Davis Medical Center, UC Davis School of Medicine, 4150 V Street, Suite 3500, Sacramento, CA 95817, USA; jzko@ucdavis.edu (J.Z.-H.K.); shjohn@ucdavis.edu (S.J.); 2Institute for Regenerative Cures, UC Davis, 2921 Stockton Blvd, Sacramento, CA 95817, USA

**Keywords:** inflammatory bowel disease, Crohn’s disease, ulcerative colitis, perianal fistula, mesenchymal stem/stromal cells, efficacy, safety

## Abstract

Inflammatory bowel disease (IBD) is a chronic inflammatory disorder of the gut that can lead to severe gastrointestinal symptoms, malnutrition, and complications such as fistulas and cancer. Mesenchymal stem/stromal cells (MSCs) are being investigated as a novel therapy for IBD and have been demonstrated to be safe and effective for perianal fistulizing Crohn’s disease (PFCD). This systematic review aims to present the most recent studies on the safety and efficacy of MSC therapy in IBD. A detailed search strategy of clinical trials on MSCs and IBD was performed on PubMed, with 32 studies selected for inclusion in this review. The newest studies on local MSC injection for PFCD continue to support long-term efficacy while maintaining a favorable safety profile. The evidence for systemic MSC infusion in luminal IBD remains mixed due to marked methodological heterogeneity and unclear safety profiles. Although further studies are needed to better establish the role of this novel treatment modality, MSCs are proving to be a very exciting addition to the limited therapies available for IBD.

## 1. Introduction

Inflammatory bowel disease (IBD) is characterized by a disordered immune response resulting in pathological inflammation in the gastrointestinal tract [1]. The two main subtypes of IBD are ulcerative colitis (UC), which affects the colon and rectum in a continuous fashion, and Crohn’s disease (CD), which can affect any part of the digestive tract from mouth to anus [1]. IBD patients can present initially with abdominal pain, diarrhea, hematochezia, and unintentional weight loss [1]. Chronic uncontrolled IBD can lead to significant complications including malnutrition, fibrostenotic strictures, formation of fistulas, intraabdominal abscesses, and colorectal cancer [1,2]. Additionally, IBD can have extraintestinal manifestations which can severely damage the skin, joints, eyes, and other organs [3].

The epidemiology of IBD has changed dramatically over the last century. In long industrialized regions such as North America and Europe, it is estimated that IBD carries an overall stable prevalence of 0.3% [2,4]. In increasingly Westernized regions such as Asia, Africa, and South America however, incidence of IBD has been rising steadily for several decades [2,4]. Given the already significant disease burden in North America and Europe and the increasing incidence in other parts of the world, IBD carries a significant economic burden globally, one that is only expected to grow moving forward [4].

Depending on the severity and extent of disease, the management of IBD can range from topical anti-inflammatory agents (e.g., 5-aminosalicylates [5-ASA]) to major abdominal surgeries for bowel resection and creation of ostomies and pouches [5,6,7]. As dysfunctional immune processes are key in the pathogenesis of IBD, one of the cornerstones of therapy is pharmacologic immunosuppression. Immunosuppression is often accomplished with immunomodulators which have been repurposed from cancer chemotherapy (e.g., 6-mercaptopurine, azathioprine [AZA], methotrexate), biologic agents which are monoclonal antibodies directed against specific cytokines, integrins, and other molecules in the inflammatory cascade, small-molecule inhibitors (e.g., tofacitinib), or some combination thereof [5,6,7,8,9].

Although immunosuppression has been demonstrated to be highly effective in IBD, loss of response in regard to symptom management and mucosal healing is known to occur in a subset of patients [10,11]. Moreover, immunosuppression is not without significant potential adverse effects (AEs) including increased risk of opportunistic infections, certain malignancies, and even paradoxical immune responses such as psoriasiform eruptions [12,13,14,15]. Additionally, these therapies, in particular biologic agents, can incur significant financial cost to the patient and the healthcare system at large [16,17].

Given the increasing prevalence of IBD and the aforementioned issues with current medical therapies, there is an unmet need to find novel treatments for IBD that are also safe and effective. Mesenchymal stem/stromal cells (MSCs) have been investigated as a novel therapy given their anti-inflammatory and tissue regenerative potential [18]. The first case of MSCs being used as treatment for IBD was published in 2003 and involved treating a woman with a rectovaginal fistula (RVF) [19]. Since then, there has been a growing body of evidence on MSC therapy showing longer term safety and efficacy for treatment of IBD [20]. MSCs are either given to patients intravenously for luminal IBD in clinical trials or through local injection into fistulous tracts for perianal fistulizing CD (PFCD) [18,20]. A meta-analysis published in 2015 showed an overall fistula healing rate of 61.3% (95% confidence interval (CI), 35.6–84.6%) for patients treated with local MSC therapy, and a remission rate of 40.5% (95% CI 7.5–78.5%) for patients treated with systemic MSC infusion [18]. Since 2015, a number of clinical trials have been published on MSC therapy in IBD, including long-term data from a phase III clinical trial which has led to the approval of Alofisel^®^ (darvadstrocel/Cx601, Takeda), a therapy composed of allogeneic adipose-derived MSCs (allo-ASCs), for PFCD in the European Union [21,22,23]. This review seeks to provide an updated overview of MSC therapy clinical trials published since 2015 with attention to longer-term efficacy and safety.

## 2. Materials and Methods 

### 2.1. Search Strategy

A systematic review of articles was performed using PubMed from inception to 29 October 2020 with the aim of updating a previously published systematic review by Dave et al. [20] with new studies. The detailed search strategy is available in Appendix A.

### 2.2. Study Selection

Studies were selected based on the following inclusion criteria: (i) human studies, (ii) included patients with IBD, (iii) MSCs or MSC-containing tissue products were used for treatment of IBD, (iv) efficacy and AEs were reported, (v) the study was published as peer-reviewed paper, letter, or abstract. Exclusion criteria were: (i) non-human studies, (ii) review papers, (iii) case reports, (iv) follow-up study available, (v) lack of results on efficacy and safety.

### 2.3. Search Results

The initial search strategy yielded 4492 abstracts for review. After initial review, 32 publications were selected for detailed review. In total, eight publications were excluded from this paper. For studies with separate publications for short-term and long-term follow-up data, only the subsequent publications with long-term follow-up data were included given they also included the short-term data, thus initial studies by Molendijk et al. [24], Panés et al. [21], and Wainstein et al. [25] will not be discussed in detail. Studies by Topal et al. [26] and Piejko et al. [27] were excluded due to the focus on cryptoglandular fistulae and the exclusion of IBD patients. A publication by Otagiri et al. [28] was excluded as it was a phase I/II study protocol without results. A study by Sanz-Baro et al. [29] was excluded as it was a retrospective study specifically examining reproductive outcomes in female patients treated with MSCs without detail on IBD outcomes. A publication by Knyazev et al. [30] was excluded due to being a retrospective analysis of several other studies on the safety of systemic MSC infusion.

In total, 24 studies were ultimately included in this review. A total of 17 studies described local administration of MSCs for PFCD; four of these studies described short-term outcomes (data collected within 6 months of treatment), while the remaining 13 studies described long-term outcomes (data collected after at least 6 months of treatment). One study which utilized both local injection and systemic administration of MSCs was included in the long-term category, as the primary type of IBD studied was PFCD [31]. The remaining seven studies described systemic administration of MSCs for luminal IBD.

## 3. Results

### 3.1. Local Injection of MSCs or MSC Containing Tissue for Perianal Fistulizing Crohn’s Disease—Short-Term Studies

A number of studies with short-term data have emerged in recent years investigating novel methods and technologies to optimize MSC therapy for IBD (Table 1). Dietz et al. [32] performed a phase I trial utilizing a bioabsorbable plug (MSC-MATRIX) impregnated with 20 million autologous adipose-derived MSCs (auto-ASCs) on 12 patients with PFCD who had failed anti-tumor necrosis factor (TNF) therapy (NCT01915927; clinicaltrials.gov). In total, 83% (10/12) achieved combined clinical and radiographic remission at 24 weeks, defined as lack of drainage upon palpation and decrease in length and diameter of the fistulous tract on T2-weighted imaging, respectively. This study also demonstrated a significant decrease in Van Assche perianal severity scores with treatment (median 13 to median 9, *p* < 0.0008). A total of two non-serious AEs related to the protocol were seroma formation at the site of fat collection. The study is limited by small sample size and lack of blinding and control group; however, the authors are planning a larger study to establish safety, efficacy, and feasibility of their MSC-MATRIX technology in PFCD. 

Lightner et al. [33] performed a phase I clinical trial using Gore Bio-A plugs coated with 35 million auto-ASCs (MSC-MATRIX) on five patients with RVF due to fistulizing CD. In prior trials studying MSCs in PFCD, patients with RVF were often excluded, thus necessitating newer trials on this patient population. At 6 months, three patients experienced complete cessation of drainage, while the other two experienced decreased drainage. None of the patients experienced radiographic remission however, as they all had persistent fistulous tracts on MRI at 6 months. No patients experienced AEs in this study, thus the authors concluded that MSC-MATRIX appears to be safe in RVFs, however efficacy could not be established. Similar to the previous study utilizing MSC-MATRIX [32], this study is limited by small sample size and lack of blinding and control group. It should also be noted that all five patients in this study underwent intestinal diversion surgery prior to enrollment, thus the findings may not be broadly applicable to the majority of patients with RVF who have not undergone intestinal diversion surgery.

**Table 1 biomolecules-11-00082-t001:** Short-term studies on local injection of mesenchymal stem/stromal cells on Perianal Crohn’s Disease. Allo-ASCs, allogeneic adipose stem cells. Auto-ASCs, autologous adipose stem cells. CD, Crohn’s disease. MSC, mesenchymal stem/stromal cells. MRI, magnetic resonance imaging.

Study	Study Type	*N*	Intervention	Primary Outcomes	Results	Comment
Dietz et al. [32]	Phase I study without blinding or control group	12 perianal CD	Implantation of bioabsorbable plug coated with 2 × 10^6^ auto-ASCs.	Healing of fistula on exam and MRI at 24 weeks.	10/12 (83%) achieved combined remission at 24 weeks.	NCT01915927 MSC-MATRIX technology
Dige et al. [34]	Phase I study without blinding or control group	21 perianal CD	Injection of freshly harvested adipose tissue around fistula tract and internal opening. Repeat injections performed if clinical healing not achieved.	Healing of fistula on exam and no fluid-conducting tract at former fistula on MRI at 6 months.	12/21 (57%) achieved clinical remission 6 months after last injection. 8/9 patient who underwent MRI had complete resolution at 6 months.	NCT03803917 Two patients required second injection. One patient required third injection.
Lightner et al. [33]	Phase I study without blinding or control group	5 rectovaginal CD	Implantation of bioabsorbable plug coated with 3.5 × 10^6^ auto-ASCs.	Healing of fistula on exam and decrease in T2 hyperintense tract on MRI at 6 months.	3/5 (60%) with complete clinical response. 2/5 (40%) with partial clinical response. 0/5 patients with radiographic remission	MSC-MATRIX technology
Nikolic et al. [35]	Phase I study without blinding or control group	4 rectovaginal CD	Single injection of 3 × 10^6^ allo-ASCs around fistula tract.	Fistula closure and absence of drainage on exam at 6 months.	1/4 (25%) achieved clinical healing at 6 months.	Darvadstrocel (Alofisel^®^) used.

Dige et al. [34] performed a prospective interventional study on 21 patients with PFCD utilizing fresh autologous adipose tissue collected on the day of injection (NCT03803917; clinicaltrials.gov). Fresh adipose tissue was utilized with the hypothesis that this may be a faster and more cost-effective alternative to in vitro expansion of ASCs which takes several weeks. This study was also notable for offering repeat injections to patients who did not achieve clinical response 6 weeks after initial treatment or who experienced later relapse on follow-up. The amount of adipose tissue injected varied depending on the length of the fistula tract (median 46 mL). Clinical healing was the outcome, defined as the patient having no symptoms of discharge, no visible external fistula opening, and no palpable internal opening on digital rectal exam. In total, 43% (9/21) achieved sustained clinical remission 6 months after single injection; 8/9 of those patients had complete resolution of fistula on pelvic MRI. A total of nine patients who did not achieve clinical remission 6 weeks after initial injection underwent a second injection; of those patients, two achieved complete clinical remission 6 months after the second injection, while another two experienced decreased fistula secretion. In total, four patients underwent a third injection of adipose tissue, of which one patient achieved clinical remission at 6 months and other experienced decreased fistula secretion. In sum, 57% (12/21) patients achieved sustained clinical remission 6 months after their last injection of fresh adipose tissue. The most common AEs were proctalgia and pain at liposuction site. A total of two patients who received larger volumes of adipose tissue (64 mL and 108 mL) developed abscesses related to the injections (ischiorectal and labial) requiring incision and drainage. In addition to lack of blinding and control group, this study is limited by the fact that pelvic MRI was not used to confirm healing in patients with anovaginal fistulas. Furthermore, there was a lower rate of fistula healing in this study after single injection (43%) with adipose tissue compared to higher fistula healing rates noted in studies that use an ASC only product administration (>55%) suggesting the advantage of using a purified ASC population for treatment.

Nikolic et al. [35] performed a small pilot study utilizing Alofisel^®^, (darvadstrocel, Takeda), a product comprised of allo-ASCs already approved for use in the European Union for PFCD, on four patients with RVF due to CD. Only one patient (25%) experienced sustained clinical healing of RVF on rectovaginal exam 6 months after treatment. The remaining three patients demonstrated open RVF at a median of 19 days. The three patients that did not respond to Alofisel^®^, had prior surgeries for CD, thus implying that disease severity and a complex fistula may have an impact on MSC efficacy. There were no intraoperative or post-operative complications, however two patients did develop abscesses requiring drainage several weeks post-treatment due to recurrent RVF. Like the other short-term studies for local injection of MSCs, this study was limited by small sample size and lack of blinding and control group. Unlike other studies on local injection of MSCs and PFCD, radiography was not used to assess healing in this study. Additionally, patients in this study were only offered single injection of 30 million ASCs, while in practice patients with PFCD can receive multiple injections of up to 120 million ASCs. 

### 3.2. Local Injection of Mesenchymal Stem/Stromal Cells for Perianal Fistulizing Crohn’s Disease—Long-Term Data

In recent years, there has been a growing body of evidence showing the long-term outcomes of MSC therapy on IBD (Table 2). In 2018, Panés et al. published long-term follow-up data of a phase III, randomized, double-blind clinical trial (ADMIRE-CD) that took place at 49 hospitals in seven European countries and Israel (NCT01541579; clinicaltrials.gov) [21,22]. In total, 212 patients with refractory PFCD were randomized 1:1 to receive either 120 million allo-ASCs (compound Cx601) or normal saline (placebo). Using modified intention-to-treat (mITT) analysis on all patients who received study treatment and had at least one post-baseline efficacy assessment, 51.5% (53/103) of patients treated with Cx601 achieved combined clinical and radiographic remission, respectively, defined as lack of suppuration on physical exam and absence of fluid collections > 2 cm, at 24 weeks compared to 35.6% in the placebo group. At 52 weeks, an even greater proportion of patients in the treatment arm achieved combined remission (56.3%) compared to the placebo group (38.6%). The most common AEs were anal fistula/abscesses and proctalgia. Limitations to ADMIRE-CD were the exclusion of patients with non-perianal fistulas (RVFs, abdominal fistulas), moderate or severely active luminal CD, and those who have had any CD-related surgeries beyond drainage and seton placement [21]. The work of Panés et al. served as the basis for darvadstrocel (Alofisel^®^) being approved for use in PFCD in the European Union [23].

Ciccocioppo et al. [36] published follow-up data on eight patients with refractory PFCD who underwent serial injections of autologous bone marrow-derived MSCs (auto-BM-MSCs) and were followed for 6 years. The mean Crohn’s Disease Activity Index (CDAI) score fell from 300 to close to 150 over the 6 years. The probability of fistula relapse-free survival declined over time, starting at 88% at 1 year, falling to 50% at 2 years, and remaining at 37% for the remainder of the 6-year follow-up. The cumulative probabilities of medication-free and surgery-free survival were 88% and 100% at year 1, 25% and 75% in years 2–4, and 25% and 63% at years 5 and 6, respectively. There were no AEs related to MSC therapy. Aside from the very small sample size, one limitation of this study is the lack of radiographic data. Though the authors cited that there may be a delay of at least 1 year between clinical healing and radiographic healing, the 6-year follow-up can more than accommodate the one-year lag.

García-Arranz et al. [37] performed a phase I/II clinical trial using allo-ASCs on ten patients with RVF due to PFCD (NCT00999115; clinicaltrials.gov). In total, 20 million cells were injected into the fistulous tract and vaginal submucosa on initial injection; if clinical healing was not achieved at 12 weeks, a second injection of 40 million cells was offered. The patients were followed up to 52 weeks after first injection. At 12 weeks, two patients achieved complete fistula healing (defined as rectovaginal re-epithelialization and absence of discharge on exam), however one of those patients was subsequently excluded from the study due to CD exacerbation requiring biologic therapy. In total, seven of the eight patients who did not demonstrate fistula healing at 12 weeks underwent second injection with 40 million cells; of those seven patients, two demonstrated complete healing at 52 weeks, another two did not, and the other three did not complete the study due to introduction of biologic therapy. Of the five total patients who completed the study, 60% (3/5) demonstrated sustained healing of their RVF at 52 weeks. No AEs related to MSC therapy were observed. Similar to other studies examining MSC therapy in RVF [33,35], this study was limited by small sample size, lack of blinding, and lack of control group. Additionally, patients in this study were required to discontinue biologic therapy prior to the study in order to prevent the benefits of biologic therapy from being interpreted as efficacy for MSC therapy. The discontinuation of biologic therapy may have led to five of the ten participants withdrawing from the study due to CD exacerbation requiring biologic therapy, thus significantly reducing this study’s sample size.

Herreros et al. [38] published data from a compassionate use program evaluating 45 patients with a total of 52 surgically refractory anal fistulae of different etiologies and their response to various types of MSC therapies including allo-ASCs, auto-ASCs, and stromal vascular fraction (SVF), the latter of which is thought to contain ASCs with minimal amounts of adipocytes and erythrocytes. In total, 18 patients had PFCD, three of whom had RVF. Patients were followed every 3 months with median follow-up time of 20 months (range 6–48 months), with assessment for clinical healing via lack of suppuration on palpation at each visit. A total of 55% (10/18) of PFCD patients achieved complete healing with a mean time of 6.5 months; 100% of PFCD patients achieved either complete or partial healing. In total, three PFCD patients injected with SVF required a second treatment to achieve complete healing, two of whom received auto-ASCs for their second treatment. Auto-ASCs led to the highest rate of complete healing (66.6%), followed by allo-ASCs (55.5%), and SVF (40%), however it is unclear from the paper the exact number of CD patients that received each treatment. It should also be noted that none of the PFCD patients with RVF achieved complete healing, only partial healing. No AEs related to MSC therapy were observed by the authors. Due to the marked heterogeneity of the patient population and the MSC therapies stemming from the compassionate use, it is overall difficult to draw concrete conclusions about MSC therapy in PFCD from this study.

Garcia-Olmo et al. [39] published a study on 10 patients with recurrent perianal fistulae of varying etiology receiving MSCs in a compassionate use program. In total, three of the patients in the study had PFCD, all of whom received auto-ASCs, two of whom received fibrin glue as well. At 1 year, 66% (2/3) of the CD patients demonstrated complete clinical healing, defined as complete re-epithelialization and lack of suppuration on exam. Interestingly, the CD patient who did not experience clinical healing had a transsphincteric fistula with anal stenosis and did not receive fibrin glue with the ASCs. No AEs related to MSC therapy were observed. Similar to Herreros et al. [38], the compassionate use paradigm leads to marked heterogeneity in the patient population and MSCs, thus limiting the ability to draw specific conclusions about MSC therapy in PFCD from this study, although it does provide data on real world use of MSCs outside a clinical trial setting.

Cho et al. [40] published 24-month follow-up data on 41 of 43 patients in a phase II trial on single injection of auto-ASCs mixed with fibrin glue, with the injection amount varying with fistula size (NCT01011244 and NCT01314079; clinicaltrials.gov). Utilizing mITT analysis on all patients with data at 24 months, 80% (28/35) experienced complete healing by physical exam (closure of tract and lack of drainage) at 1 year, and 75% (27/36) continued to experience complete healing at 2 years. In total, 53 AEs were observed in 30 patients, the most common of which were abdominal pain, eczema, CD exacerbation, anal inflammation, diarrhea, and fever; no AEs directly related to ASCs were observed. It should be noted that this study lacks radiographic data, thus no conclusions can be made about combined healing when comparing to other studies.

Park et al. [41] performed an open-label, dose escalation study on six patients with PFCD utilizing a single injection of allo-ASCs and fibrin glue. The first group of three patients received 1 × 10^7^ cells/mL (group 1); four weeks after determining the initial dose was safe, a second group of three patients received 3 × 10^7^ cells/mL (group 2). The final injection volume depended on the fistula length, with group 1 receiving on average 4.33 × 10^7^ cells, and group 2 receiving 1.7 × 10^8^ cells on average. In total, 50% of patients (two in group 1 and one in group 2) experienced complete clinical healing, defined as closure of external opening and lack of drainage and inflammation on inspection, at 8 months after single injection. Regarding AEs, five patients experienced post-procedural pain. One patient was hospitalized due to development of a new fistula; however, this was deemed unrelated to MSC therapy. Due to the very small sample size, the study could not define an ideal dose for MSCs. This study also lacks radiographic data, which may also limit this study’s applicability compared to other studies with data on combined healing.

In 2018, Wainstein et al. [42] published long-term follow-up data on a short-term study originally published in 2016 [25] on nine patients with refractory PFCD who first underwent seton placement followed 4–6 weeks afterwards by closure of internal fistula opening with a small endorectal flap and injection of 100–120 million auto-ASCs with platelet-rich plasma (PRP). The rationale behind including PRP in the treatment is that it would have beneficial effects on wound healing, neovascularization, and may even serve as a chemoattractant for MSCs [43]. Between the nine patients, there were 11 fistulae. The patients were followed for a median of 31 months (range 21–37 months), and at the end of the follow-up, 89% (8/9) of the patients and 91% (10/11) of the fistulae demonstrated complete healing via physical exam (complete epithelialization and absence of suppuration) without relapse. No AEs related to MSC/PRP therapy were observed in this study. Limitations of this study include small sample size and lack of radiographic data.

Zhou et al. [44] performed an open-label clinical trial on 22 patients with PFCD who were randomized 1:1 to receive either curettage and seton-placement followed by auto-ASC injection or a control procedure of incision-thread-drawing (China Clinical Trials Registry No. ChiCTR1800014599). Primary endpoints were combined clinical (complete epithelialization of external opening) and radiographic (absence of fistula on MRI) healing. At 3, 6, and 12 months, the combined healing proportion for the experimental group vs. the control group were 90.9% (10/11) vs. 45.5% (5/11), 72.7% (8/11) vs. 54.5% (6/11), and 63.6% (7/11) vs. 54.5% (6/11), respectively, however statistical significance was not achieved at any timepoint. The most common AE in both groups was perianal pain. A total of five patients (two in the experimental group and three in the control group) underwent operations due to perianal disease recurrence, and thus they were removed from the study. No serious AEs (SAEs) were observed in the study. The authors note that while their auto-ASC protocol does appear effective, time and cost may be prohibitive, as patients do have to undergo two separate procedures (liposuction and MSC injection), with an intervening period of several weeks to allow for in vitro expansion of MSCs. 

Knyazev et al. [31] published a study utilizing both local injection and systemic administration of allo-BM-MSCs in patients with PFCD. Patients were divided into three groups: group 1 (*n* = 12) underwent local and systemic MSC therapy, group 2 (*n* = 10) received infliximab, and group 3 (*n* = 14) received unspecified antibiotics and immunosuppressants. Patients in group 1 received local injections of 40 million MSCs into the fistulous tract at weeks 0, 4, and 8, and systemic infusions of 150–200 million MSCs at weeks 0, 1–2, 12, and 52. At 3 months and 6 months, clinical healing of fistulae were noted at 66.6% (8/12) in group 1, 60% (6/10) in group 2, and 7.14% (1/14) in group 3. At 1-year follow-up, clinical healing declined slightly in group 1 to 58.3% (7/12), was preserved in group 2 at 60%, and rose slightly in group 3 to 14.3% (2/14). At 2 years, clinical healing declined in all groups to 41.6% (5/12) in group 1, 40% (4/10) in group 2, and 0% (0/14) in group 3. No safety profile was provided in this study. Additionally, it is unclear what component of MSC therapy (local injection or infusion) contributed to efficacy, as these two components were not administered separately in this study. The authors concluded that combined local and systemic MSC therapy and infliximab were both superior to antibiotics and immunosuppressants for PFCD, however efficacy was noted to decline in all groups after 2 years.

Serrero et al. [45] performed a small phase I clinical trial on 10 patients with treatment-refractory PFCD injected with a combination of autologous microfat and SVF, with the hypothesis that the trophic and volumizing effects of microfat would promote tissue healing. Combined clinical remission by physical exam (absence of suppuration at external opening) and radiographic remission by MRI (absence of collection > 2 cm) were evaluated at 12 and 48 weeks. At 12 weeks, 70% of patients had clinical response, but only 20% achieved combined remission; the proportions grew by 48 weeks, when 60% of patients achieved combined remission with 80% achieving clinical response. Non-serious AEs included pain at lipoaspiration site and skin reaction due to anesthetics. A total of three SAEs occurred: two CD flares and one new fistula, however, it is unclear whether these AEs were related to the study protocol. The study is limited by small sample size and by being unblinded and uncontrolled. 

Laureti et al. [46] published a prospective pilot study on 15 patients with medically refractory PFCD utilizing the Lipogems^®^ device to process autologous adipose tissue (NCT03555773; clinicaltrials.gov). Lipogems^®^ is a device used to rapidly process adipose tissue through reduction of adipose tissue clusters and elimination of oily substances and blood residues, resulting in microfragmented adipose tissue and preserved SVF, similar to the treatment in Serrero et al. [45]. After curettage and fistulectomy, 20cc of the processed adipose tissue was injected into the internal fistula opening and fistula tract. At 24 weeks, 10 patients demonstrated combined clinical (closure of external opening on exam) and radiographic (absence of fluid collection > 3mm on pelvic MRI) remission; an additional four patients achieved clinical remission, while only one patient did not respond to treatment. At 24 months, these results were maintained without relapse. In total, three patients who did not achieve combined remission had anal stenoses prior to treatment. Regarding AEs, 20% (3/5) experienced subcutaneous hematoma from liposuction, and one patient developed perianal bleeding requiring suture placement. Similar to Dige et al. [34], the authors expressed optimism in utilizing freshly harvested autologous adipose tissue for use in PFCD, however this study is limited by small sample size, lack of control group, and lack of blinding. 

Barnhoorn et al. [47] published long-term follow-up data from a study by Molendijk et al. [23] in 2015 which was a randomized, double-blind, placebo-controlled, dose-finding study utilizing allogeneic bone marrow-derived MSCs (allo-BM-MSCs) in 21 patients with medically refractory PFCD (NCT01144962; clinicaltrials.gov). After seton removal, curettage, and closure of the internal fistula tract, patients were randomized to either injection of placebo solution (*n* = 6), 10 million (cohort 1, *n* = 5), 30 million (cohort 2, *n* = 5), or 90 million MSCs (cohort 3, *n* = 5) to the fistula tract. At 12 weeks, 80% (4/5) patients in cohort 2 achieved the primary endpoint of combined clinical (absence of discharge and closure of external opening) and radiographic fistula (improvement of fluid in fistulae compared to initial MRI) healing; at 4 years, 100% (4/4) patients demonstrated persistent clinical healing (one patient was lost to follow-up). Only 33% (2/6) of patients in the placebo group achieved fistula healing at 12 and 24 weeks; in the long-term follow-up period, two patients received BM-MSCs and one patient received Cx601 (darvadstrocel), while the remaining three patients did not demonstrate any clinical healing. In cohort 1, fistula healing rose from 40% (2/5) at 12 weeks to 80% (4/5) at 24 weeks; with 75% (3/4) demonstrating persistent clinical healing at 4 years (one patient died of a cecal adenocarcinoma). In cohort 3, only 20% (1/5) achieved fistula healing at all timepoints. Short-term, all patients reported 1 week of post-procedural anal pain and bloody/purulent discharge, and one patient in each group developed a perianal abscess requiring drainage. As noted, one patient in cohort 1 was diagnosed with cecal adenocarcinoma 15 months after MSC therapy; given the patient had a strong family history of colorectal cancer (CRC) at younger ages (patient’s uncle died of CRC at age 42), this was thought unrelated to MSC therapy. At 4 years, six patients in the treatment cohorts developed perianal abscesses, three experienced active CD, and five patients were treated for infections. One patient in cohort 2 developed B-cell lymphoproliferative disease (LPD) in the rectum, but after short tandem repeat analysis of the BM-MSCs and the LPD tissue, the LPD was thought unlikely to be related to the MSC therapy [48]. This study highlights the need for long term follow up to ascertain the safety of MSC for PFCD.

**Table 2 biomolecules-11-00082-t002:** Long-term studies on local injection of mesenchymal stem/stromal cells for Perianal Crohn’s Disease. Allo-ASCs, allogeneic adipose stem cells. Allo-BM-MSCs, allogeneic bone marrow mesenchymal stem/stromal cells. Auto-ASCs, autologous adipose stem cells. CD, Crohn’s disease. mITT, modified intention-to-treat. MSC, mesenchymal stem/stromal cells. MRI, magnetic resonance imaging. SVF, stromal vascular fraction.

Study	Study Type	*N*	Intervention	Primary Outcomes	Results	Comment
Cho et al. [40]	Phase II	41 perianal CD	Single injection of fibrin glue and 3 × 10^7^ auto-ASCs/cm of fistula length.	Complete closure of fistula tract on exam at 24 months.	At 24 months, complete healing was observed in 21/26 (80.8%) patients in mPP group and 27/36 (75%) in mITT group.	NCT01011244 & NCT01314079
Ciccoccioppo et al. [36]	Phase I	10 perianal CD	Serial intrafistular injections of auto-BM-MSCs every 4 weeks (median 4 injections)	CDAI, healing by exam and MRI, and fistula relapse-free survival yearly for 6 years.	Mean CDAI score decreased from 300 to 150 at 6 years. Fistula relapse-free survival was 88% at 1 year, 50% at 2 years, and remaining at 37% for the remainder of the six-year follow-up.	
Garcia-Olmo et al. [39]	Phase I	10 (3 with Perianal CD)	Single injection of auto-ASCs in the CD patients, 2 of whom also received fibrin glue.	Complete healing: re-epithelialization and absence of suppuration at 1 year.	2/3 (66%) of CD patients achieved complete healing at 1 year.	Study on both CD and non-CD fistulae from compassionate use program.
Park et al. [41]	Phase I	6 perianal CD	Group 1 (*n* = 3): 10^7^ cells/mL and fibrin glue. Group 2 (*n* = 3): 3 × 10^7^ cells/mL and fibrin glue.	Complete healing on physical exam and MRI at 8 months.	2/3 (66.7%) in group 1 and 1/3 (33.3%) in group 2 achieved clinical healing at 8 months.	
García-Arranz et al. [37]	Phase I/II	10 rectovaginal CD	Single injection of 2 × 10^6^ allo-ASCs. If clinical healing not achieved at 12 weeks, second injection of 4 × 10^6^ cells.	Complete healing: re-epithelialization of both vaginal and rectal sides and absence of drainage at 52 weeks.	Of 5 total patients who completed study, 3 achieved clinical healing (60%) at 52 weeks.	NCT00999115 7 patients underwent second injection.
Panés et al. [22]	Phase III	212 perianal CD	Injection of 1.2 × 10^8^ allo-ASCs (*n* = 107) vs. placebo (*n* = 105).	Combined remission: absence of external openings on exam and absence of collections > 2 cm on MRI at week 52.	Using mITT, 58/103 (56.3%) of patients in treatment arm achieved combined remission vs. 39/101 (38.6%) in control arm at week 52.	NCT01541579Long-term data for ADMIRE-CD trial [21] Led to approval of darvadstrocel in European Union.
Wainstein et al. [42]	Phase I	9 perianal CD	Seton placement followed 4–6 weeks later by endorectal flap and injection of 1–1.2 × 10^8^ auto-ASCS with platelet-rich plasma.	Complete healing: absence of suppuration from the external fistula opening and complete epithelialization. Partial healing: external fistula opening remaining open, but with a decrease of > 50% in suppuration and size of the external fistula opening.	8/9 (88.9%) patients with complete healing, 1/9 (11.1%) patients with partial healing at median follow-up of 31 months.	Long-term study for Wainstein et al. [41]
Knyazev et al. [31]	Phase II	36 Perianal CD	Group 1 (*n* = 12) local injection of 4 × 10^7^ allo-BM-MSCs at weeks 0, 4, and 8, as well as systemic infusion of 1.5–2 × 10^8^ allo-BM-MSCs at weeks 0, 1–2, 12, and 52. Group 2 (*n* = 10) received infliximab. Group 3 (*n* = 14) received antibiotics and immunosuppressants.	Healing by epithelialization, decrease in suppuration and discomfort.	At 3 and 6 months, healing of 66.6% (8/12) in group 1, 60% (6/10) in group 2, and 7.14% (1/14) in group 3. At 1-year, healing of 58.3% (7/12) in group 1, 60% (6/10) in group 2, and 14.3% (2/14) in group 3. At 2 years, healing 41.6% (5/12) in group 1, 40% (4/10) in group 2, and 0% (0/14) in group 3.	
Herreros et al. [38]	Phase I/II	45 (18 with perianal CD)	Injection of allo-ASCs, auto-ASCs, or SVF. 5 patients underwent second injection.	Complete healing: absence of suppuration.	55.5% of CD patients achieved healing. 40% of CD patients who received SVF achieved healing. 66.6% of CD patients who received auto-ASCs achieved healing. 55.5% of CD patients who received allo-ASCs achieved healing.	Study on both CD and non-CD fistulae from compassionate use program.
Barnhoorn et al. [47]	Phase I	21 perianal CD	Group 1 (*n* = 5): 10^7^ allo-BM-MSCs. Group 2 (*n* = 5): 3 × 10^7^ allo-BM-MSCs. Group 3 (*n* = 5): 9 × 10^7^ allo-BM-MSCs. Group 4 (*n* = 6): placebo.	Healing of fistula on exam and MRI at 4 years.	Group 1: 3/4 (75%) healing at 4 years. Group 2: 4/4 (100%) healing at 4 years. Group 3: 2/5 (20%) healing at 4 years. Group 4: 0/3 (0%) healing at 4 years.	NCT01144962 Long-term study for Molendijk et al. [24] 1 patient lost to follow-up in groups 1 and 2. 3 patients lost to follow-up in group 4.
Zhou et al. [44]	Phase II	22 perianal CD	Control: incision-thread-drawing. Treatment: seton followed by auto-ASC injection 2 weeks later.	Closure of fistulae by exam and MRI at 3, 6, and 12 months.	3 months: 10/11 (91%) in treatment arm vs. 5/11 (45.5%) in placebo arm. 6 months: 8/11 (72.7%) in treatment arm vs. 6/11 (54.5%) in placebo arm. 12 months: 7/11 (63.6%) in treatment arm vs. 6/11 (54.5%) in placebo arm.	China Clinical Trials Registry No. ChiCTR1800014599
Laureti et al. [46]	Phase I	15 perianal CD	Single injection of microfragmented adipose tissue processed with Lipogems^®^ system.	Combined remission: closure of all external openings on exam and absence of collections > 3mm on MRI at 24 months.	10/15 (66.7%) with combined remission at 24 months. 14/15 (93.3%) with only clinical remission at 24 months.	NCT03555773 Lipogems^®^ system utilized.

### 3.3. Systemic Mesenchymal Stem/Stromal Cell Therapy for Luminal Inflammatory Bowel Disease

There have been several studies in recent years on the emerging use of systemic MSC therapy on IBD (Table 3). Dhere et al. [49] performed a phase I safety trial on auto-BM-MSCs infusion in 12 patients with moderate-severe CD, defined as CDAI > 220 and failure of immunomodulators and/or biologics for at least 3 months during disease course (NCT01659762; clinicaltrials.gov). A total of four patients were randomized to each of the three treatment groups: low dose (2 × 10^6^ cells/kg), intermediate dose (5 × 10^6^ cells/kg), and high dose (1 × 10^7^ cells/kg). In total, 42% (5/12) experienced clinical response 2 weeks after infusion as defined by reduction in CDAI of over 100 points. In total, two patients experienced AEs thought to be related to treatment: one patient in the low dose group developed acute appendicitis 9 days after infusion, and one patient in the intermediate dose group developed *Clostridioides difficile* colitis 30 days after infusion. Due to small sample size and very short follow-up time, larger and longer-term studies on auto-BM-MSC infusion are necessary to truly establish efficacy moving forward.

Knyazev et al. [50] published a study utilizing systemic infusion of allo-BM-MSCs in patients with acute UC (newly diagnosed within the past 6 months). The experimental group (*n* = 12) received “standard anti-inflammatory therapy” consisting of a 5-ASA preparation and corticosteroid taper in addition to allo-BM-MSCs at a dose of 1.5–2 million cells/kg at weeks 0, 1, and 26, while the control group (*n* = 10) received “standard anti-inflammatory therapy” alone. Using the Truelove and Witts severity index, 58.3% (7/12) of experimental group patients and 60% (6/10) of control patients had severe UC. At 1-year follow-up, the remission rates, recurrence rates, and clinical and endoscopic indices did not differ between the two groups. At 2-year follow-up, the average duration of remission differed in a statistically significant manner, with the average duration in the experimental group being 22 months, and the average duration in the control group being 17 months (*p* = 0.049). Additionally, the recurrence rate was three times lower in the experimental group than in the control group at 2 years (*p* = 0.03). At 3 years, the duration of remission was similar between the two groups (22 months in the experimental group, 20 months in the control group), however UC endoscopic severity was more severe in the control group when compared to the experimental group (Rakhmilewitz Index 8.1 vs. 4.75, respectively, *p* < 0.001). No safety profile was provided in the paper. Given initial disease severity varied within groups, it would be prudent to further examine whether initial disease severity affects response to MSC therapy. The authors concluded that although systemic MSC therapy increased efficacy of standard anti-inflammatory therapies in acute UC, regular administration may be necessary for maintenance.

Knyazev et al. [51] published a study investigating for a possible synergistic effect of adding AZA to systemic allo-BM-MSC infusion in patients with luminal CD. In total, 15 patients in group 1 with an average CDAI of 337.6 received MSC therapy with 2–2.5 mg/kg of AZA, while 19 patients in group 2 with an average CDAI of 332.7 received MSC therapy alone. All patients received 2 million MSCs/kg at weeks 0 and 26. At 2 months, all patients in both groups achieved clinical remission (CDAI < 150), with no statistically significant difference in CDAI scores between the two groups. In fact, the average CDAI scores at 6 months and 12 months were below 150 in both groups with no statistically significant difference. When measuring cytokine levels, the authors note significantly lower levels of interferon γ, TNFα, and interleukin 1β in the group that received AZA. No safety profiled was provided in this paper. Based on the cytokine profiles, the authors concluded that the addition of azathioprine to infusion of MSCs may produce a more robust anti-inflammatory effect than MSCs alone, however given the clinical remission rates were similar in both groups, it is unclear how much of an effect MSC therapy actually had in this study.

Gregoire et al. [52] published a letter-to-the-editor on an open-label phase I/II trial on the intravenous administration of allo-BM-MSCs in 13 patients with severe CD, defined as CDAI 220–450, C-reactive protein > 5 mg/L, fecal calprotectin > 150 μg/g, refractory to standard medical therapies (NCT01540292; clinicaltrials.gov). Patients were treated with two injections of 1.5–2.0×10^6^ MSCs/kg at weeks 0 and 4. Although eight patients had decreases in CDAI by week 8, only 2/13 (15%) met the primary endpoint of CDAI decrease of over 100 points at that timepoint, and only 1/13 (7.7%) achieved clinical remission (CDAI < 150). One patient developed a mild upper respiratory tract infection requiring antibiotics, otherwise no infusion reactions or other AEs were observed. This study is limited by its small sample size and lack of blinding and control group, thus larger studies will be needed to both establish optimal dosing and frequency elucidate efficacy.

Melmed et al. [53] published findings from a phase Ib/IIa study examining infusion of a preparation of placenta-derived mesenchymal-like adherent cells (PDA-001) in patients with moderate-severe CD, defined as CDAI 220–450 and endoscopic evidence of inflammation within three months of study (NCT01155362; clinicaltrials.gov). In the open-label phase Ib study, four patients received eight units (1.2 × 10^9^ cells) of PDA-001 at each of two infusions set 1 week apart. In phase Ib, one patient experienced a grade 3 hypersensitivity reaction, while another patient experienced venous thrombosis at the infusion site. Due to the AEs in phase Ib, it was determined that patients would be randomized 1:1:1 to either placebo (*n* = 16), 1 unit (*n* = 15), or 4 units (initial *n* = 15, however two patients excluded from study, making final *n* = 13) of PDA-001 in the double-blind phase IIa study. A total of five patients in each treatment group achieved the primary endpoint of clinical response with CDAI decrease of over 100 points and/or 25% from baseline by weeks 4 and 6, compared to 0% in the placebo group (*p* < 0.05). Additionally, two patients in each treatment group achieved clinical remission (CDAI < 150) compared to none in the placebo group. A possible limitation to the study is the relatively long disease duration (mean 10+ years) in each group with over half the subjects having been exposed to anti-TNF therapy, thus the study population may overly represent more medically refractory CD. The most common treatment-related AEs in treated patients were erythema, pyrexia, and headache. A total of three patients in the study experienced SAEs thought to be treatment related: hypersensitivity reaction, gastric ulcer perforation, and anal cancer. Due to the two SAEs that occurred in another study on PDA-001 in rheumatoid arthritis (myocardial infarction, retinal artery spasm), this study was halted early prior to the enrollment of the two final patients.

Hu et al. [54] performed a phase I/II randomized clinical trial evaluating infusion of umbilical cord MSCs (Um-MSCs) on patients with moderate-severe UC defined as Mayo UC activity score 8–10 ((NCT01221428; clinicaltrials.gov). In total, 34 patients were randomized to receive two infusions of Um-MSCs (once intravenously, then via superior mesenteric artery seven days later), while 36 patients received normal saline in the same fashion. The primary outcome was clinical response defined as decrease in total Mayo UC activity score of ≥3 and ≥30% from baseline, with accompanying decrease in rectal bleeding subscore of ≥1 point or absolute subscore for rectal bleeding of 0 or 1. At 3 months, 29/34 (85.3%) group I vs. 6/36 (16.7%) achieved clinical response, with sustained responses documented up to 24 months. No treatment-related AEs were noted in the publication. Although the data appear promising for Um-MSCs, lack of randomized assignment, lack of blinding, and similar levels of biomarkers (CRP and ESR) between the two groups are major drawbacks of the study.

**Table 3 biomolecules-11-00082-t003:** Systemic administration of mesenchymal stem/stromal cells for luminal inflammatory bowel disease. 5-ASA, 5-aminosalicylic acid. Allo-BM-MSCs, allogeneic bone marrow mesenchymal stem/stromal cells. AZA, azathioprine. CD, Crohn’s disease. CDAI, Crohn’s Disease Activity Index. HBI, Harvey-Bradshaw Index. SAEs, serious adverse events. UC, ulcerative colitis. Um-MSCs, umbilical cord mesenchymal stem/stromal cells.

Study	Study Type	*N*	Intervention	Primary Outcomes	Results	Comments
Melmed et al. [53]	Phase Ib/IIa	50 luminal CD	Phase Ib: two infusions of 8U PDA-001 (1.5 × 10^9^ cells) one week apart. Phase IIa: two infusions of placebo, 1U PDA-001 (1.5 × 10^8^ cells), or 4U PDA-001 (6 × 10^8^ cells) one week apart.	Decrease in CDAI by ≥100 points and/or 25% from baseline at weeks 4 and 6.	Phase Ib: primary efficacy not reported Phase IIb: Placebo: 0/16 1U PDA-001: 5/15 (33%) 4U PDA-001: 5/13 (38.5%)	NCT01155362 PDA-001 is comprised of allogeneic placental MSCs. Study was suspended early due to several SAEs.
Dhere et al. [49]	Phase I	12 luminal CD	Single infusion of 2, 5, or 10 × 10^6^ auto-BM-MSCs/kg.	Decrease in CDAI by ≥ 100 points at 2 weeks.	5/11 (45.4%) achieved clinical response.	NCT01659762
Hu et al. [54]	Phase I/II	70 with UC	Group I: IV injection of 0.5 × 10^6^ Um-MSCs/kg, followed by intra-arterial injection of 1.5 × 10^7^ MSCs one week later. Group II: placebo (normal saline) in same manner as group I.	Decrease in total Mayo UC activity score of ≥3 and ≥30% from baseline, with accompanying decrease in rectal bleeding subscore of ≥1 point or absolute subscore for rectal bleeding of 0 or 1.	29/34 (85.3%) with clinical response in group I vs. 6/36 (16.7%) at 3 months.	NCT01221428
Knyazev et al. [50]	Phase I	22 with UC	Control: 5-ASA and steroid taper. Treatment: 1.5–2 × 10^6^ allo-BM-MSCs/kg at weeks 0, 1, and 26, in addition to 5-ASA and steroid taper.	Remission rate and average remission duration.	Remission rate of 50% (6/12) in treatment group vs. 10% (1/10) in control group at 3 years. Remission duration of 22 months in treatment group vs. 20 months in control group at 3 years.	
Gregoire et al. [52]	Phase I/II	13 luminal CD	2 injections of 1.5–2.0 × 10^6^ allo-BM-MSCs /kg 4 weeks apart.	Decrease in CDAI by ≥100 points at 8 weeks.	2/13 (15.4%) with clinical response at 8 weeks.	NCT01540292
Zhang et al. [55]	Phase I	82 luminal CD	Control: “background treatment.” Treatment: infusion of 1 × 10^6^ Um-MSCs/kg once a week for 4 weeks.	Decrease in CDAI, HBI, and corticosteroid usage.	CDAI decreased by 62.5 in Um-MSC vs. 23.6 in control at 12 months. HBI decreased by 3.4 in Um-MSC vs. 1.2 in control at 12 months. Corticosteroid dosage decreased by 4.2 mg/day in Um-MSC vs. 1.2 mg/day in control at 12 months.	NCT02445547
Knyazev et al. [51]	Phase I/II	34 luminal CD	Group 1: 2 × 10^6^ allo-BM-MSCs/kg at months 0, 1, and 6, with AZA 2–2.5 mg/kg. Group 2: 2 × 10^6^ allo-BM-MSCs/kg at months 0, 1, and 6.	Clinical remission (CDAI < 150) at 12 months.	At 12 months, average CDAI was 99.9 in group 1, 100.6 in group 2.	

Zhang et al. [55] performed a prospective, randomized, controlled, open-label clinical trial utilizing Um-MSCs on 82 patients with moderate–severe CD (CDAI 220–450) who were on steroid maintenance therapy for 6 months or more (NCT02445547; clinicaltrials.gov). In total, 41 patients were randomized to receive four weekly infusions of 1 × 10^6^ cells/kg Um-MSCs, while the other 41 patients continued “background treatment.” For the treatment group at 12 months, the CDAI was 62.5 points lower than at the start of the study, whereas the CDAI only fell by 23.6 points in the control group. Similarly, the Harvey-Bradshaw Index (HBI) fell by 3.4 points in the treatment group by 12 months, whereas the HBI was only lower by 1.2 points in the control group. All patients who remained in the study underwent colonoscopy at 12 months, and the Crohn’s Disease Endoscopy Index of Severity fell from 9.2 to 3.4 in the treatment group, compared to a decrease from 8.7 to 7.5 at 12 months in the control group. No patients in either group achieved clinical remission (CDAI < 150) throughout the study. A total of four patients in the treatment group developed fevers with infusion that resolved with symptom control. Additionally, seven patients in the treatment group developed a total of nine URIs within 6 months of infusion. No SAEs were noted in the study. In addition to being an open-label study, the study is limited by the fact that “background treatment” in the control group was not explicitly defined.

## 4. Discussion

The number of studies exploring MSC therapy in IBD has grown substantially in recent years. The evidence is particularly strong for local injection of ASCs for PFCD, for which there now exists a medication (Alofisel^®^/darvadstrocel, Takeda) approved for use for this indication in the European Union [23]. The safety profile for local injection also appears favorable as well, as the most common AEs were peri-procedural proctalgia, and in some instances abscess. MSC therapy carries a theoretical risk of malignancy, however this has not borne out thus far from the existing studies. Of note, per Sanz-Baro et al. [29], MSC therapy does not appear to affect fertility and pregnancy outcomes in the limited number of pregnant women who underwent this treatment in their study. The safety profile of local MSC therapy will be further elucidated in the coming years as more long-term data become available and more patients are treated in Europe as part of routine clinical care.

One condition in which local MSC therapy has not been established to be consistently effective is RVF [33,35,37]. It has been hypothesized that RVF is particularly difficult to treat due to the shortness of the tracts and thinness of the rectovaginal membrane, thus overall decreasing the volume and amount of MSCs that can be delivered relative to non-rectovaginal fistulae [33,35]. More studies focusing specifically on RVFs and their idiosyncrasies are necessary moving forward.

There is an increasing number of studies which seek to further optimize local MSC therapy in a variety of ways. Studies utilizing autologous MSCs required in vitro expansion of MSCs for several weeks prior to administration, thus adding time to the patient’s care [40,42,44]. As reviewed above, newer studies by Laureti et al. [46] and Dige et al. [34] seek to use freshly processed adipose tissue that can be injected on the day of collection using novel processing methods, however, these studies have a much lower efficacy compared to a purified MSC product. Multiple investigators are also using adjunctive methodologies to augment the effectiveness of MSCs, i.e., to develop “next-generation” MSCs. The use of “next-generation” MSC therapies may involve pretreatment with small molecules and cytokines, genetic engineering of MSCs to deliver bioactive factors to damaged tissues [56], or combining them with bioabsorbable plugs (MSC-MATRIX) or scaffolds as a novel means to deliver MSCs [32,33]. Further studies are needed to see if these modifications to MSC therapy will increase efficiency while preserving or even improving upon safety and efficacy.

The evidence for systemic infusion of MSCs in IBD remains mixed due to marked methodological heterogeneity between studies, compounded by lack of evidence showing MSCs reaching the intestine after intravenous injection and unclear safety profiles. As reported in this review, systemic infusions of allo-BM-MSCs, Um-MSCs, and placental MSCs have all been trialed in IBD. Placental MSCs were deemed unsafe to the point of suspension of the clinical trial for CD, as the three SAEs documented in that study (hypersensitivity reaction, gastric ulcer perforation, anal cancer) were deemed possibly related to the therapy, and a concurrent study examining placental MSCs in rheumatoid arthritis led to a retinal artery occlusion and a myocardial infarction [53]. The Um-MSC studies originating from China appear promising, however larger studies will be necessary to further substantiate the benefits [54,55]. The data on BM-MSCs also appears varied, with single-center studies by Knyazev et al. [31,50,51] reporting significant benefit, but studies by other investigators outside of Russia with more subdued results [49,52]. Although the safety of BM-MSC infusion is reported to be safe by Knyazev et al. [30], this study is again limited by the data originating from a single center. Further studies with more uniform methodologies and more descriptive safety profiles are needed in order for progress to be made in systemic infusions of MSCs for IBD.

Our systematic review has potential limitations. Non-human studies were not included in this review, as the aim of the review was to describe relevant clinical outcomes in humans. Initial studies that had subsequent publication of long-term follow-up data were excluded, thus decreasing the number of short-term studies discussed in this review; this was purposefully done as the follow-up publications were felt to have sufficiently included and described the short-term data. Furthermore, we only included studies published from 2015 in this systematic review. While there are a number of studies published prior to 2015 on MSC therapy in IBD, these were mostly early phase studies given the relative nascency of MSC therapy at that time and have been comprehensively described in our previous systematic review [18]. Thus, for this systematic review we chose to focus on recent studies that were more likely to be methodologically consistent and contain important longer-term efficacy and safety data, especially as MSC therapy becomes increasingly available for IBD patients in clinical practice.

In conclusion, recent studies on local MSC injection for PFCD continue to support long-term efficacy while maintaining a favorable safety profile. The evidence for systemic MSC infusion in luminal IBD remains mixed due to marked methodological heterogeneity, and unclear safety profiles. Although further studies are needed to better establish the role of this novel treatment modality, MSCs are proving to be a very exciting addition to our therapeutic armamentarium for IBD.

## 5. Conclusions

IBD is a chronic inflammatory condition of the digestive tract that can lead to abdominal pain, diarrhea, hematochezia, and complications such as perianal fistulas, all of which can significantly impact a patient’s quality of life. MSCs have been investigated as a novel therapy for IBD due to their immunomodulatory and anti-inflammatory effects. There exists robust evidence for the efficacy and safety of local injections of MSCs in PFCD, to the point where darvadstrocel (Alofisel^®^, Takeda), comprised of allo-ASCs, has been approved in the European Union for use in PFCD. Studies with the goal of further optimizing the efficiency and effectiveness of local MSC therapy are ongoing with the investigation of novel techniques such as use of fresh adipose tissue, co-injection with adjunctive agents, and use of bioabsorbable plugs. The data and safety profiles emerging from studies evaluating systemic infusion of MSCs in luminal IBD suggest safety but equivocal efficacy, however. While further studies are necessary to fully establish efficacy and safety, MSCs are emerging to be a very promising and exciting addition to the armamentarium of therapies available for IBD.

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
