# Peer review of "Efficacy and Safety of Mesenchymal Stem/Stromal Cell Therapy for Inflammatory Bowel Diseases: An Up-to-Date Systematic Review"

_biomolecules, 2021, doi:10.3390/biom11010082_

Round 1
Reviewer 1 Report
In the manuscript “Efficacy and safety of mesenchymal stem/stromal cell therapy for inflammatory bowel diseases: an up-to date systematic review”, authors provided an updated review of clinical trials of MSC therapy with a focus on efficacy and safety.
With the growing body of data on the use of cell therapy methods to treat a variety of diseases, such reviews are highly relevant. They make it easier for both clinicians and researchers to access and expand research around the world and to bring the development of safe innovative medicines into a routine clinic.
The review presents the results of 32 studies published in the last 5 years.
The review was written professionally, classified according to short-term and long-term outcome results, compared MSCs of various origins, the effectiveness of the appointment method, and analyzed in other sections. With the growing body of data on the use of cell therapy methods to treat a variety of diseases, such reviews are highly relevant. They facilitate access to and expansion of research around the world by clinicians and researchers, and the development of safe innovative medicines into a conventional clinic.
The manuscript deserves to be published. Nevertheless I have 1 major and 1 minor remark.
Major remark:
All analyzes were performed according to published results for phase I, I / II or II (and even 1 phase III) clinical trials. But there was no any information on the registration of these studies. Authors have to add information on which of the analyzed studies are registered in the clinical trial databases and to add the registration numbers.
Minor remark:
In the section 2.3 Search Results (line 87-88 ), authors declared: "For studies with separate publications for short-term and long-term data, only long-term data were included." Nevertheless, the safety and efficacy of MSCs in short-term studies was analyzed in the manuscript ( Section 3.1. Local Injection of MSCs or MSC containing tissue for Perianal Fistulizing Crohn’s Disease – Short-term Studies).
I want authors to explain the difference between excluded short-term studies and those analyzed in the manuscript.
This information will make this review a little better.
My final decision is – this manuscript can certainly be published with some mentioned corrections
Author Response
We appreciate the meaningful comments provided by the journal reviewers to help improve the quality of our manuscript. In addition to making minor grammatical edits and formatting changes to the tables in order to fit the journal template, we have addressed the comments point by point and incorporated the changes in the revised manuscript as per journal guidelines (view “All Markup” under Tracking).
Reviewer #1:
Major remark:
All analyzes were performed according to published results for phase I, I / II or II (and even 1 phase III) clinical trials. But there was no any information on the registration of these studies. Authors have to add information on which of the analyzed studies are registered in the clinical trial databases and to add the registration numbers.
Author Response: Thank you for this very important feedback. We added clinical trial database registration numbers to the text and tables when available.
Minor remark:
In the section 2.3 Search Results (line 87-88 ), authors declared: "For studies with separate publications for short-term and long-term data, only long-term data were included." Nevertheless, the safety and efficacy of MSCs in short-term studies was analyzed in the manuscript ( Section 3.1. Local Injection of MSCs or MSC containing tissue for Perianal Fistulizing Crohn’s Disease – Short-term Studies).
I want authors to explain the difference between excluded short-term studies and those analyzed in the manuscript.
Author Response: We recognize this point needed clarification. We further described why certain short-term studies with subsequent publications on longer-term data were excluded (lines 90-93). In sum, for initial short-term studies for which a subsequent article was published containing long-term outcomes, the decision was made to focus on the subsequent publication given that it would also sufficiently include and describe the short-term data. Additionally, an aim of this systematic review was on longer-term clinical outcomes given the increasing availability of studies with long-term efficacy and safety data. The short-term studies that were fully included in this review focused on novel methods to optimize MSC therapy in IBD; as this was important to discuss in the context of future directions of MSC therapy.
Reviewer 2 Report
General comment: Review article entitled “Efficacy and safety of mesenchymal stem/stromal cell therapy for inflammatory bowel diseases: an up-to date systematic review” presents the recent scientific data about novel approaches for IBM therapy. This is a well-organized study, which cover the majority of the aspect relevant to the specific issue. Some minor corrections are required for the improvement of the manuscript.
Abstract: The Abstract is well written and adequately presents the theoretical background and the aim of the review article.
-Could authors add 1-2 lines about the basic findings of the review study?
Introduction: The introduction section is well-written and covers the importance to further investigate new approaches for the therapy of IBM.
-Authors should divide the introduction session into paragraphs.
Main text: Authors adequately and analytically present the recent studies about the specific issue. The recent studies are summarized and well presented in useful tables.
-Material and methods: Could authors further explain, mainly on discussion session, why chose only 24 studies from a greater number of initial studies? Could the inclusion criteria not to be so limited?
-Could authors shortly discuss possible limitations of the study?
References: The references used by the authors cover adequately the relative scientific field and the aims of the study.
Author Response
We appreciate the meaningful comments provided by the journal reviewers to help improve the quality of our manuscript. In addition to making minor grammatical edits and formatting changes to the tables in order to fit the journal template, we have addressed the comments point by point and incorporated the changes in the revised manuscript as per journal guidelines (view “All Markup” under Tracking).
Abstract: The Abstract is well written and adequately presents the theoretical background and the aim of the review article.
-Could authors add 1-2 lines about the basic findings of the review study?
Author Response: Thank you for this important feedback. We added two sentences to the abstract summarizing the basic findings of the systematic review (lines 18 – 21): “The newest studies on local MSC injection for PFCD continue to support long-term efficacy while maintaining a favorable safety profile. The evidence for systemic MSC infusion in luminal IBD remains mixed due to marked methodological heterogeneity and unclear safety profiles.”
Introduction: The introduction section is well-written and covers the importance to further investigate new approaches for the therapy of IBM.
-Authors should divide the introduction session into paragraphs.
Author Response: Thank you for the suggestion. We have divided the introduction into paragraphs.
Main text: Authors adequately and analytically present the recent studies about the specific issue. The recent studies are summarized and well presented in useful tables.
-Material and methods: Could authors further explain, mainly on discussion session, why chose only 24 studies from a greater number of initial studies? Could the inclusion criteria not to be so limited?
-Could authors shortly discuss possible limitations of the study?
Author Response: Thank you for this important feedback. We recognize a more thorough discussion on inclusion/exclusion and possible limitations was necessary for this review to be more complete. Thus, we added a paragraph to the end of the discussion section regarding these points.
"Our systematic review has potential limitations. Non-human studies were not included in this review, as the aim of the review was to describe relevant clinical outcomes in humans. Initial studies that had subsequent publication of long-term follow-up data were excluded, thus decreasing the number of short-term studies discussed in this review; this was purposefully done as the follow-up publications were felt to have sufficiently included and described the short-term data. Furthermore, we only included studies published from 2015 in this systematic review. While there are a number of studies published prior to 2015 on MSC therapy in IBD, these were mostly early phase studies given the relative nascency of MSC therapy at that time and have been comprehensively described in our previous systematic review[18]. Thus, for this systematic review we chose to focus on recent studies that were more likely to be methodologically consistent and contain important longer-term efficacy and safety data, especially as MSC therapy becomes increasingly available for IBD patients in clinical practice."